# Farey tree and devil's staircase of frequency-locked breathers in ultrafast lasers

Xiuqi Wu[1], Ying Zhang[1], Junsong Peng [1,2] ✉, Sonia Boscolo[3], Christophe Finot[4] & Heping Zeng [1,5,6] ✉

Nonlinear systems with two competing frequencies show locking or resonances. In lasers, the two interacting frequencies can be the cavity repetition rate and a frequency externally applied to the system. Conversely, the excitation of breather oscillations in lasers naturally triggers a second characteristic frequency in the system, therefore showing competition between the cavity repetition rate and the breathing frequency. Yet, the link between breathing solitons and frequency locking is missing. Here we demonstrate frequency locking at Farey fractions of a breather laser. The winding numbers exhibit the hierarchy of the Farey tree and the structure of a devil's staircase. Numerical simulations of a discrete laser model confirm the experimental findings. The breather laser may therefore serve as a simple test bed to explore ubiquitous synchronization dynamics of nonlinear systems. The locked breathing frequencies feature a high signal-to-noise ratio and can give rise to dense radio-frequency combs, which are attractive for applications.

Nonlinear systems with two competing frequencies show locking or resonances, in which the system locks into a resonant periodic response featuring a rational frequency ratio[1]. The locking increases with nonlinearity, and at subcritical values of the nonlinearity, the system has quasi-periodic responses between locked states, whilst the supercritical system may exhibit chaotic as well as periodic or quasi-periodic responses. A general feature of frequency locking is the robustness of the locked states to variations of system parameters, namely, the constancy of the frequency ratio (or winding number) over a range of parameters. Frequency locking has been investigated theoretically and experimentally in many physical systems including coupled oscillators[2], charge-density waves[3], Josephson junctions[4,5], and the Van der Pol oscillator[6] amongst others[7], and their distribution in parameter space in the form of a devil's staircase[8] is well understood from the number theory concept of Farey trees[9–14]. In optics, frequency-locking phenomena have been extensively studied in modulated semiconductor lasers, where an external frequency can be readily coupled to the nonlinear system by using a radio-frequency

(RF) source[11,15–18], and the hierarchy of the Farey tree and structure of a devil's staircase can be rather easily observed when tuning the external frequency[11]. Frequency locking has also been demonstrated in other laser structures, such as fibre lasers with external loss modulation[19] or solid-state lasers operating in a two-mode regime[20]. Furthermore, although not explicitly mentioned by the authors, the subharmonic, harmonic, and rational harmonic operation regimes of Kerr microresonators that were reported in refs. 21,22 imply a frequency-locking process. The generation of soliton molecules (i.e., stable bound states of two solitons) in a titanium-sapphire laser that was reported in ref. 23 also evidences the occurrence of frequency locking: a subharmonic response of the soliton molecule was observed when the strength of the external driving force exceeded a certain threshold.

All the frequency-locking examples mentioned above relate to nonlinear systems where an external, accurately controllable modulation provides a new frequency to the system. Far less is known, by comparison, when the second frequency is not externally controlled and is intrinsic to the nonlinear system. This is particularly relevant to

[1]State Key Laboratory of Precision Spectroscopy, East China Normal University, 200241 Shanghai, China. [2]Collaborative Innovation Center of Extreme Optics, Shanxi University, 030006 Taiyuan, Shanxi, China. [3]Aston Institute of Photonic Technologies, Aston University, Birmingham B4 7ET, UK. [4]Laboratoire Interdisciplinaire Carnot de Bourgogne, UMR 6303 CNRS—Université de Bourgogne Franche-Comté, F-21078 Dijon, Cedex, France. [5]Chongqing Key Laboratory of Precision Optics, Chongqing Institute of East China Normal University, 401120 Chongqing, China. [6]Shanghai Research Center for Quantum Sciences, 201315 Shanghai, China. ✉e-mail: jspeng@lps.ecnu.edu.cn; hpzeng@phy.ecnu.edu.cn

breathing solitons that have recently emerged as a universal ultrashort pulse regime in passively mode-locked fibre lasers[24–28]. Breathing solitons are localised structures showing periodic variations in their parameters. They are found in various subfields of natural science, such as solid-state physics, fluid dynamics, plasma physics, chemistry, molecular biology and nonlinear optics[29]. In optics, breathers were first observed in Kerr fibre cavities[30] and subsequently found in microresonators[21,31,32]. They are currently attracting significant research interest in virtue of their relationship with a rich set of important nonlinear phenomena, such as rogue wave formation[33,34], the Fermi–Pasta–Ulam recurrence[35–37], turbulence[38], chimera states[39,40], chaos[41] and modulation instability phenomena[42]. From a practical application perspective, breathers contribute to improving the accuracy of dual-comb spectroscopy[43] as the breathing frequency comes along with additional tones in a frequency comb, and the breather laser can produce strong ultrashort pulses without using compressors[44,45].

In this paper, we present the first in-depth study of the locking of breather oscillations to the cavity repetition frequency in a fibre laser. Besides the hurdle represented by the absence of an external driver to realise frequency locking, the excitation of breathing solitons requires fine-tuning of the laser parameters, as the parameter space of breathing soliton mode locking is much narrower than stationary mode locking[44]. Therefore, targeting frequency-locked breather states in the laser via trial and error is a laborious task. Here we show that such a difficulty can be circumvented by using an evolutionary algorithm (EA). Machine-learning methods, referring to using statistical techniques and numerical algorithms to carry out tasks without explicit programmed and procedural instructions, are widely deployed in many areas of engineering and science[46]. In the field of ultrafast photonics, machine-learning approaches and the application of genetic and evolutionary algorithms have recently led to several dramatic improvements in dealing with the multivariable optimisation problem associated with reaching desired operating regimes in fibre lasers. In the present study, the merit function used in the EA optimisation procedure can distinguish between frequency-locked and unlocked breather states, thereby enabling fast and precise tuning of the laser to the target frequency-locked breather operation. The locked breather states show two unambiguous features: persistence under pump power and polarisation perturbations, and narrow linewidth and high signal-to-noise ratio (SNR) of the oscillation frequency in the electrical spectrum of the laser emission. Importantly, frequency-locked states occur in the order they appear in the Farey tree and within a pump-power range equalling the width of the corresponding step in the devil's staircase. This demonstrates that breather mode-locked fibre lasers exhibit the ubiquitous competing dynamics of two interacting frequencies in coupled systems.

## Results

### Frequency-locked and unlocked breathers in the laser

To investigate the dynamics of breathers, we have built the fibre ring cavity that is sketched in Fig. 1a. Pump light is provided by a 980-nm laser diode and it is delivered to the unidirectional cavity through a wavelength-division multiplexer. A piece of erbium-doped fibre (1.25 m) is employed as the gain medium. Other fibres in the cavity are standard single-mode fibres. These two optical fibres have group-velocity dispersion (GVD) parameters of 65, and −22.8 ps²/km, respectively, resulting in a net cavity dispersion of 0.009 ps² around 1565 nm. The laser has a repetition rate ($f_r$) of 34.2 MHz. Mode locking is realised through an effective saturable absorber by the nonlinear polarisation evolution (NPE) effect[47]. The transfer function of the saturable absorber (NPE) is controlled via three wave plates based on liquid crystal (LC) phase retarders working together with a polarisation beam splitter (PBS). The PBS is also used as an output coupler. The emitted light from the laser is monitored by several diagnostic systems. A portion is measured directly by a 50-GHz photodiode (PD1) connected to a real-time oscilloscope with a bandwidth of 33 GHz and a sampling rate of 80 GSa/s. The second laser output is input to a dispersive Fourier transform (DFT) setup which constitutes a long segment fibre that cumulates a GVD large enough for the stretched waveform to represent the spectral intensity of the initial pulse waveform[48]. Thus, the optical spectrum of each pulse can be measured in real time by the oscilloscope through detecting the output signal from the DFT setup (PD2). Additional measurement devices are used to characterise the spectral properties of the laser output: an optical spectrum analyser, an electrical spectrum analyser (ESA), and a cymometer.

Breathing solitons can be excited by tuning the pump strength and the cavity loss (polarisation controllers)[24] in the laser. Figure 2a, b

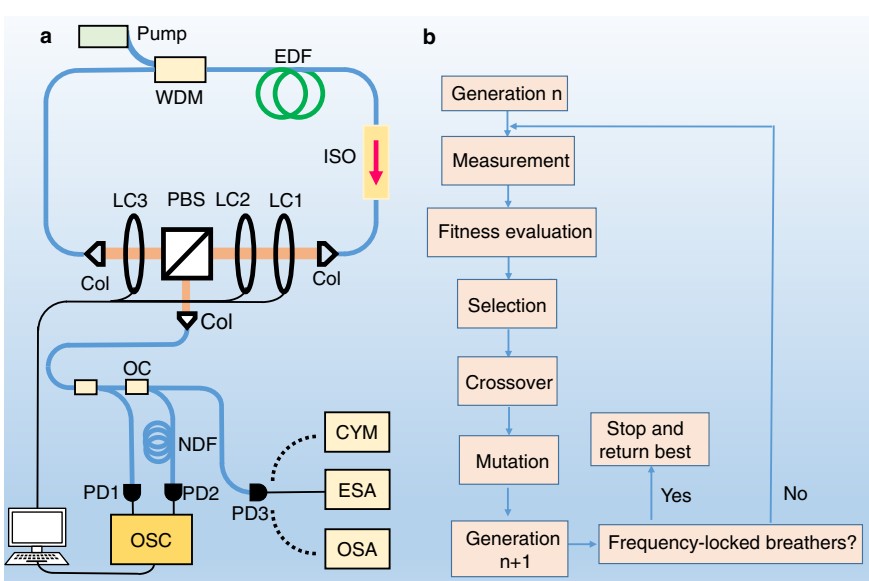

**Fig. 1 | The experimental configuration. a** The breather fibre laser setup. WDM wavelength-division multiplexer, EDF erbium-doped fibre, ISO isolator, LC liquid crystal phase retarder, PBS polarisation beam splitter, Col collimator, OC optical coupler, NDF normally dispersive single-mode fibre involved in the dispersive Fourier transform measurements, PD photodetector, OSC oscilloscope, CYM cymometer, ESA electronic spectrum analyser, OSA optical spectrum analyser. **b** The flowchart of the evolutionary algorithm.

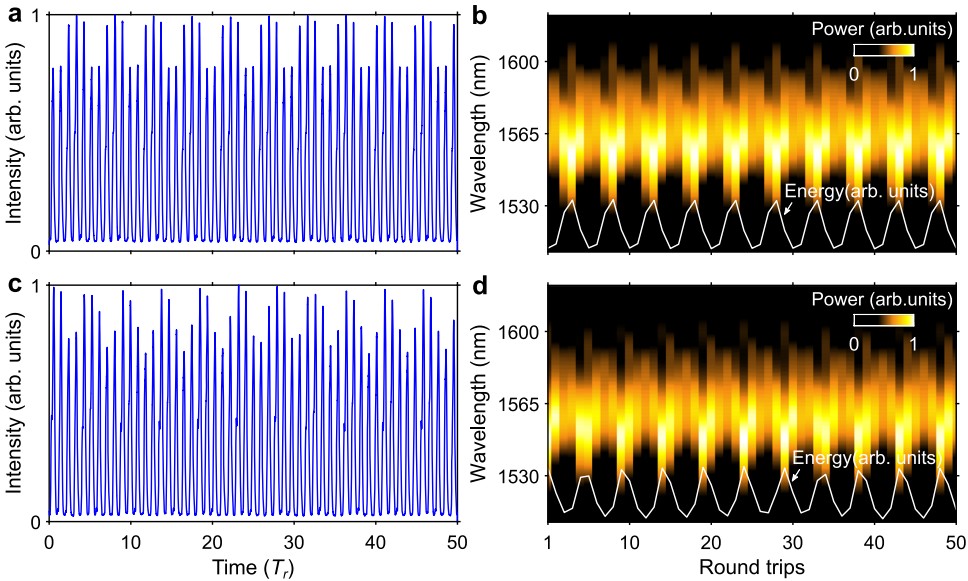

**Fig. 2 | Two different breather operations of the laser observed over 50 cavity roundtrips. a, b** Frequency-locked breather state showing a well-defined periodicity, and **c, d** frequency-unlocked breather state featuring degraded periodicity. Panels **a, c** show the photo-detected DFT (dispersive Fourier transform) output signals ($T_r$ is the round-trip time), and panels **b, d** are the corresponding DFT recordings of single-shot spectra. The white curves in **b, d** represent the energy evolutions.

shows an example of a breather operation of the laser recorded at a pump power of 74 mW. In sharp contrast with soliton pulse shaping which generates uniform pulse trains, the train of output pulses shows periodic variations in intensity occurring, in the example of Fig. 2a, across a well-defined period of 50 cavity roundtrips. Note that while Fig. 2a shows the photo-detected signal after time stretching, the same periodic evolution is also observed for the pulse train directly detected at the laser output. The corresponding spatiospectral representation of the laser regime (Fig. 2b) depicts that the optical spectrum of the pulse broadens and compresses periodically over cavity roundtrips, synchronised with the periodic variations in pulse energy (white line), which is a distinctive feature of breathing solitons. Variations in the system parameters may give rise to a different breather state in the laser as shown in Fig. 2c, d, where the pump power is decreased to 73 mW: whilst the period of oscillation seems to be unchanged, the quality of the periodic behaviour is clearly degraded in comparison with the previous case. The RF spectra of the laser emission taken from the ESA (Fig. 3) reveal the major difference between the two types of breather states. The breathing frequency of the unstable breather state shown in Fig. 2c exhibits a noisy and broad structure (Fig. 3c, d). By contrast, the stable breather state of Fig. 2a features a neat breathing frequency with narrow linewidth (0.5 Hz; see Supplementary Fig. 1 for details of the measurement) and high SNR (Fig. 3a, b). The measurements taken with the cymometer confirm the different stability properties of the breathing frequency for the two states (Fig. 3e). The breathing frequency ($f_b$) of the stable breather state is 6.84 MHz exactly equalling one-fifth of the fundamental repetition frequency, hence corresponding to a rational winding number of $f_b/f_r = 1/5$. As discussed later in this paper, this locked breathing frequency remains unchanged over a range of pump power values.

## Evolutionary algorithm optimisation of frequency-locked breathers

Reaching a frequency-locked breather state in our laser depends on precisely adjusting four parameters: the pump strength and three polarisation controllers, which is quite difficult to do manually. In ref. 49, we have introduced an approach based on an EA to search and control the breather mode-locking state in ultrafast fibre lasers. It relies on detecting and controlling the key parameter of breathers—the breathing frequency, to tune the period and breathing ratio of

breathers. In the self-tuning regime, the operation state of the laser is characterised in real time with the oscilloscope, which is linked to a computer running the EA and controlling the polarisation state through the voltages applied on the LCs via a driver to lock the system to the desired breather regime (Fig. 1a, b). Yet, the merit function of the breather mode locking used in ref. 49 is unable to distinguish between frequency-locked and unlocked breather states, where it usually breeds unlocked (unstable) states which have a wider parameter space. Here, we further develop our approach to directly pinpoint frequency-locked breathers so that the EA tunes the laser to these states only. To this end, we define a new merit function that takes into account the distinguishing trait of frequency-locked breather states, namely, a high SNR of the breathing frequency as shown in Fig. 3a, b. The new merit function is given in Eq. (3) in "Methods". Figure 4a shows a representative optimisation curve (referring to a breather state with a winding number of 1/5). It depicts the evolution of the maximum and mean merit scores of the population over successive generations along with the corresponding evolution of the SNR of the breathing frequency. We can see that the SNR grows rapidly and converges to a maximum after eight generations, thus indicating the establishment of a frequency-locked operation mode of the laser. The best merit score features a similar evolution. The measurements of the breathing frequency under pump power and ploarisation tuning shown in Fig. 4b and c, respectively, confirm that the laser works in the target regime. The reliability of the merit function of the frequency-locked breather state has been assessed by repeating the optimisation procedure numerous times, with the results showing that each time the SNR of the breathing frequency is high-frequency locking occurs. Additional examples of optimisation curves (for breather states with the winding numbers 1/5 and 2/9) are given in Supplementary Fig. 2.

## Farey tree and devil's staircase of the breather laser

Benefiting from a reliable and efficient EA-based optimisation approach, we have explored the transitions between the different breather states of the laser that can be accessed by varying the pump power starting from the range corresponding to a 1/5 frequency-locked state. Figure 5a shows an example of a plot of the breathing frequency as a function of the pump power, revealing the presence of various plateaux (steps). The spectral measurements carried out with the ESA allow us to unambiguously relate the breathing frequencies

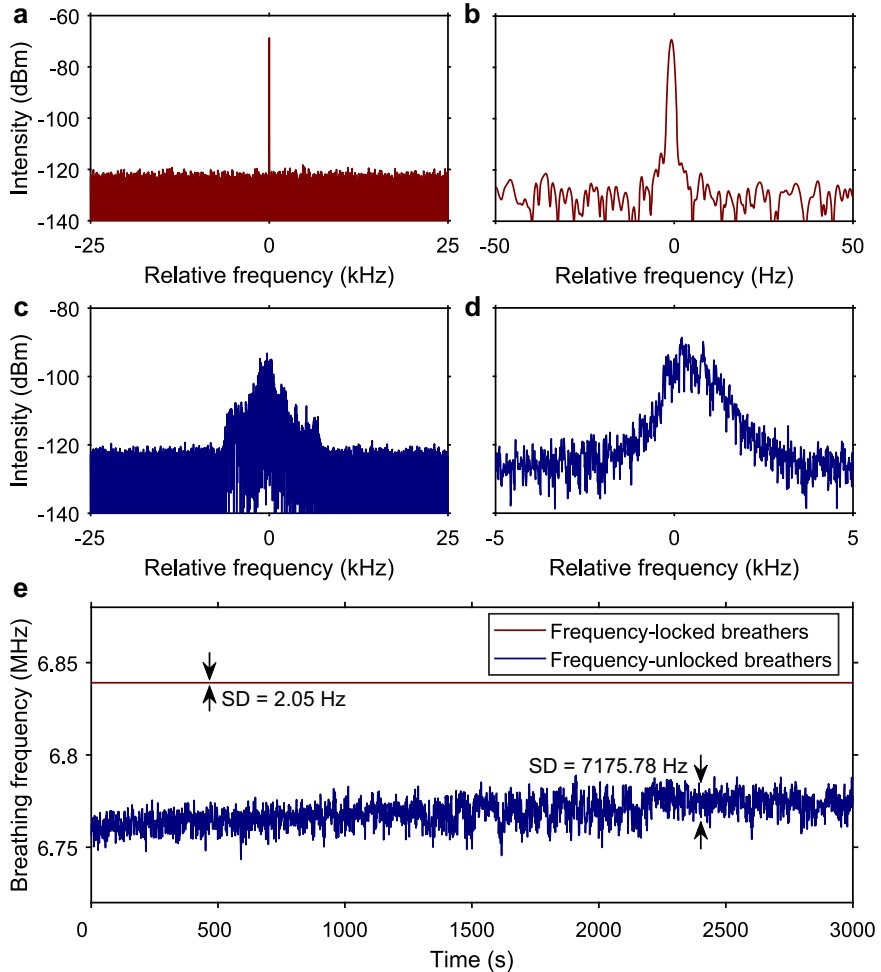

**Fig. 3 | RF spectral measurements of the breather states shown in Fig. 2.**
The reference frequency is one-fifth of the fundamental repetition frequency.
**a**, **b** Single-mode oscillation of the breathing frequency when frequency
locking occurs measured over spans of 50 kHz and 100 Hz, respectively.
**c**, **d** Unstable multimode oscillation of the breathing frequency measured
over 50-kHz and 10-kHz spans. **e** Change in breathing frequency over time
for the locked (red) and unlocked (blue) breather states, as measured with
a cymometer. The standard deviation (SD) of the breathing frequency
values is 2.05 Hz for the frequency-locked state and 7175.78 Hz for the
unlocked state.

associated with the plateaux to rational winding numbers: as shown in
Fig. 5b–d, when the laser operates in a frequency-locked state, the RF
spectrum features a finite number $n$ of spectral lines below the cavity
repetition frequency $f_r$ and equally spaced by $f_r/n$. For example, in Fig.
5d, the frequency-locked breather regime brings about the excitation
of a RF comb that is 41 times denser than that obtained when the laser
operates in the usual single-pulse stationary regime. The most intense
line in the spectrum is the breathing frequency $f_b$, and if this is the $m$th
line from the short-frequency side, then the corresponding winding
number is given by $m/n$. The temporal and spectral dynamics of the
breather patterns belonging to the winding numbers 2/9 and 9/41 are
given in Supplementary Fig. 3.

Importantly, in Fig. 5a the winding numbers occur from left to
right in the sequence predicted by the Farey tree, as shown in the inset
of the figure, and the width of the step associated with a $m/n$
frequency-locked state depends on the level where $m/n$ appears in the
Farey tree's hierarchy. The gaps (in pump power) between the stairs
(plateaux) refer to quasi-periodic breather oscillations similar to the
example shown in Figs. 2c, d and 3c, d. The fractal dimension $D$ of the
stairs can be calculated from the set of gaps (see Methods), and is
determined to be $D = 0.906 \pm 0.025$, approaching the value of 0.87
expected from a complete devil's staircase[9]. Note that fractal dimen-
sions of $0.890 \pm 0.001$ and $0.91 \pm 0.03$ were reported in refs. 8,11,
respectively. Here, the small deviation (4%) from 0.87 partly results

from the minimum power increment of the pump laser diode
(0.1 mW). The fact that the steps associated with the winding numbers
7/32 and 9/41 consist of only one point in Fig. 5a is also due to this
limitation, thus stressing the need for very robust control of the sys-
tem's properties. The process of formation of the devil's staircase is
reversible: by decreasing the pump power, nearly the same staircase
can be observed. We emphasise that contrarily to modulated external-
cavity semiconductor lasers where the modulation frequency can be
arbitrarily set hence the frequency-locked states appearing in the
order predicted by the Farey tree can be easily accessed[11], in a
breathing soliton mode-locked laser the breathing frequency is
established once the laser is fabricated, while it can be entrained by
tuning the laser parameters. Nevertheless, the Farey tree and devil's
staircase can still be observed, indicating the universality of this fractal
phenomenon. Setting the laser to a slightly different initial polarisation
state, Farey fractions belonging to other two parts of the Farey tree can
be identified through the RF spectra while tuning the pump power (see
Supplementary Figs. 4 and 5). In both cases, the calculated dimension
of the stairs approaches that of a complete devil's staircase.

Figure 5e illustrates the build-up phase of frequency locking.
Starting from a pump power of 69 mW, three radiofrequencies are
present, namely the breathing frequency ($f_b$), the difference frequency
between $f_r$ and 5th harmonic of $f_b$ ($f_r - 5f_b$), and the difference fre-
quency between the first two ($6f_b - f_r$). As $f_r - 5f_b$ approaches zero,

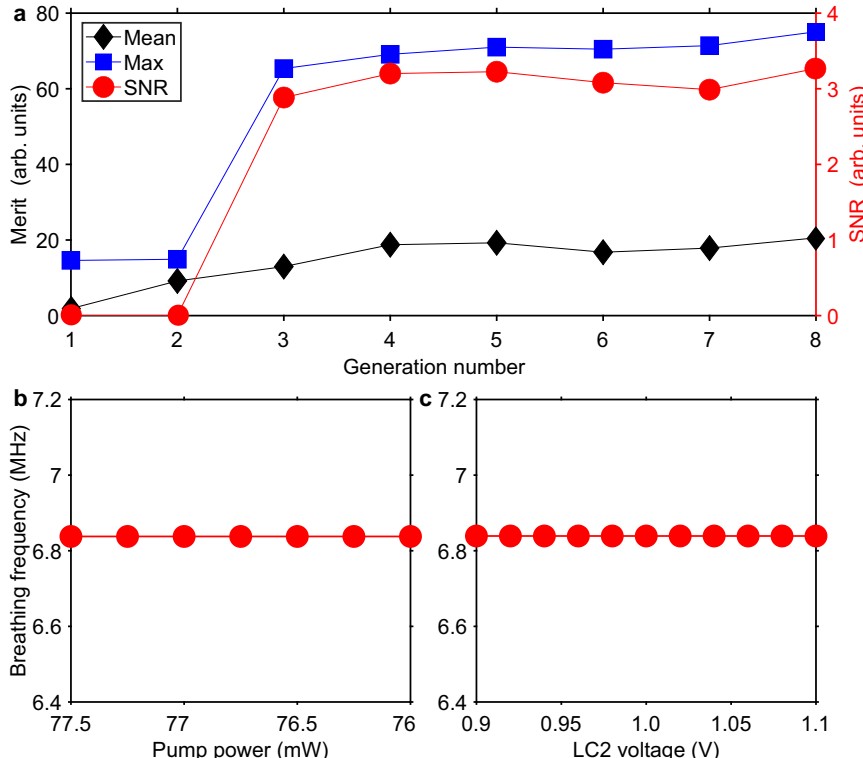

**Fig. 4 | The machine-learning results. a** Evolution of the mean (black diamonds) and best (blue squares) merit scores of the individuals for each generation. Also shown is the corresponding evolution of the SNR (signal-to-noise ratio) of the breathing frequency (red circles). Persistence of the optimal state (frequency-locked breathers) with a variation of the **b** pump power and **c** polarisation (varied by changing the voltage on the liquid crystal LC2).

frequency locking occurs at the winding number 1/5. This winding number then experiences redshifts under pump-power increments, generating other winding numbers. The map shown in Fig. 5e also evidences the very different spectral features of frequency-locked and quasi-periodic breather states. Therefore, even though the winding numbers 7/32 and 9/41 display only one point in Fig. 5a, they can clearly be identified in this map, which reveals a richness of detail that has been largely overlooked in previous studies due to the lack of high-quality RF spectral measurements. It is also noteworthy that changing the pump power by only 10% is enough to find seven frequency-locked states for the laser, whose power-stability properties are dictated by a devil's staircase. As a further note, we would like to emphasise that the frequency-locked states observed are reproducible but not self-starting, meaning that if the pump power is turned off when the laser operates in a locked state and then it is turned back on again, the laser does not return to that state instantaneously. To restore the frequency-locked operation, one can run the EA controlling the polarisation states again, which will quickly reset the laser to the desired state. Many such experimental tests have confirmed the reproducibility of the locked states.

To validate our experimental findings, we have performed numerical simulations of the laser using a scalar-field, lumped model that includes the dominant physical effects of the system on the evolution of a pulse over one round trip inside the cavity, namely, GVD and self-phase modulation for all the fibres, gain saturation and bandwidth-limited gain for the active fibre[50], and the discrete effects of a saturable absorber element (see "Methods"). The gain saturation energy in the model is related to the pump power in the experiment. Figure 6a, b shows plots of the breathing frequency (winding number) as a function of the gain saturation energy when the latter is varied starting from the range corresponding to a 1/5 locked state with a step of 10 and 1 pJ, respectively. With the smaller step, more plateaux are observed, thus confirming the fractal structure of the winding number distribution. It

is seen in Fig. 6b that the model can reproduce the same part of the Farey tree from a breathing frequency of 1/5 to 2/9 as that observed in the experiment (Fig. 5a). The gaps (in gain saturation energy) between the stairs also resemble those (in pump power) found in the experiment. The fractal dimension of the set of gaps calculated from the model is $0.873 \pm 0.09$, which is closer to the value expected from a complete devil's staircase than the experimentally calculated value because the step in gain saturation energy can be made arbitrarily small in the model. Figure 6c illustrates the build-up phase of frequency locking, which again shows good agreement with the experimental results (Fig. 5e). As mentioned above, a small change in the initial polarisation state of the laser can trigger Farey fractions belonging to a different part of the Farey tree. This experimental observation is confirmed by the results shown in Supplementary Fig. 6, which have been obtained by slightly changing the linear intracavity loss in the model.

## Discussion

We have demonstrated for the first time that a fibre laser working in the breathing soliton generation regime is a nonlinear system showing frequency locking at Farey fractions. The frequency-locked breather states of the laser are characterised by robustness against parameter (pump power and polarisation) variations and a breathing frequency with narrow linewidth and high SNR. We have exploited the latter feature to realise intelligent control of the frequency-locking process, where the use of an EA with a locked breather-tailored merit function has been the key to the precise excitation of these breather states. Indeed, contrary to previous frequency-locking demonstrations in optics relying on an external modulation applied to the system, we have been able to manipulate the intrinsic breathing frequency of the laser system. Both the experiments and simulations show that the fractal dimension of the winding numbers of breathers approaches that of devil's staircase, indicating the

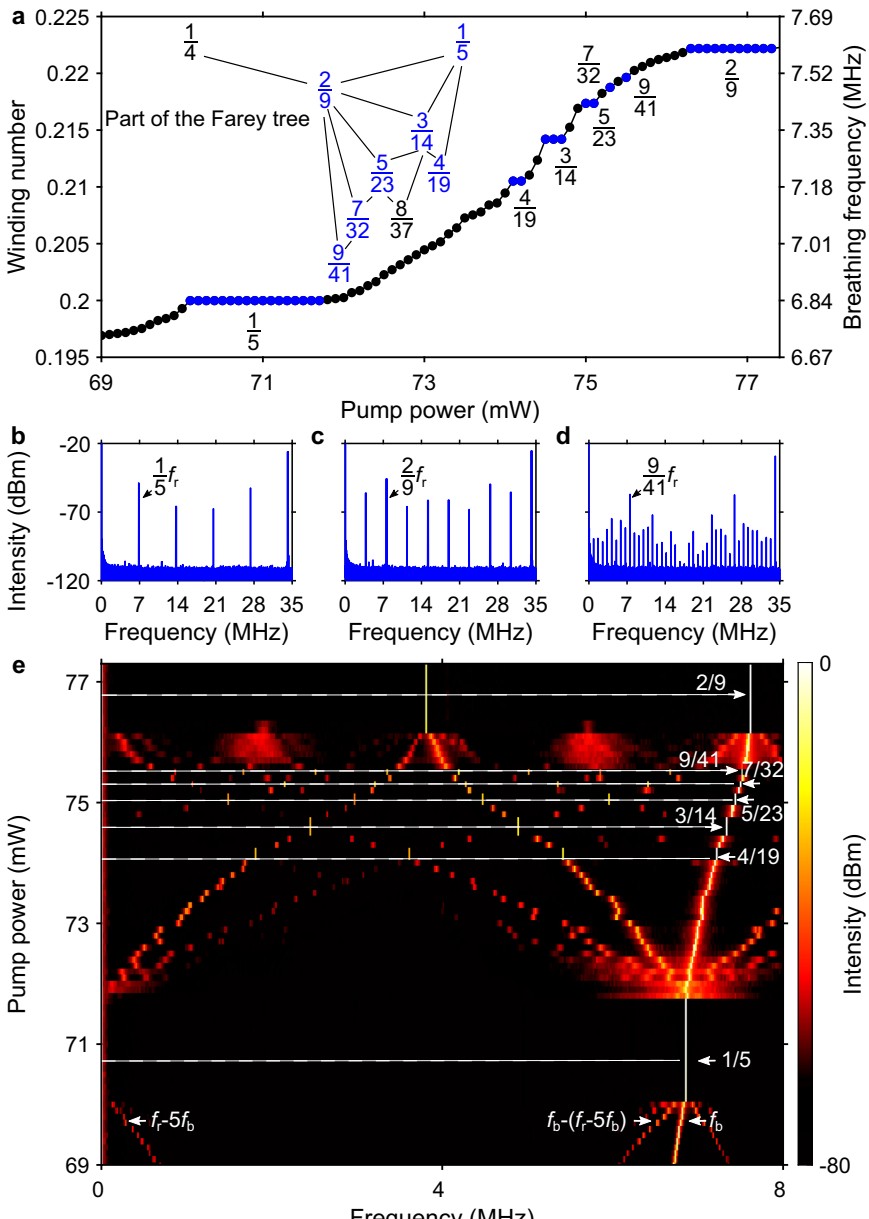

**Fig. 5 | Farey tree and devil's staircase measured in experiments. a** Measured breathing frequency (winding number) as a function of the pump power. In the inset is shown the part of the Farey tree containing the observed winding numbers (blue). **b–d** RF spectra measured with the ESA showing dense frequency combs for the frequency-locked states corresponding to the winding numbers 1/5, 2/9 and 9/ 41, respectively. A new set of equidistant spectral lines fills in the frequency interval corresponding to the cavity repetition rate $f_r$ (34.2 MHz). **e** Map of spectral intensity in the space of radio-frequency and pump power, showing the build-up of rational winding numbers.

universal nature of the laser. The breather mode-locked fibre laser thus may serve as a simple model system to investigate the universal dynamics of frequency locking. Besides, our work may stimulate the study of frequency locking in other physical systems where breathing solitons are found, where frequency locking could give a new angle on the dynamics of these systems. The EA approach used in this paper could benefit the control of the frequency-locking process in such systems as well as in others. We also believe that our EA-based approach for the control of frequency locking in fibre lasers is not restricted to NPE-based configurations and can be extended to other laser mode-locking schemes that entail period multiplication, such as the Mamyshev oscillator[51,52].

Optical breathing solitons have been extensively studied in open-loop nonlinear systems such as single-pass fibre systems[36,53]. However, in the absence of a frequency-locking mechanism, these breathers may suffer from instabilities originating from the noise of the input light. By contrast, we have studied the dynamics of breathers in a closed-loop system—a laser resonator. In this system, the universal frequency-locking process is tailored through the nonlinear interaction between the cavity repetition frequency provided by the laser resonator and the breathing frequency. Ergo, frequency-locked breathers can be generated, showing excellent stability against cavity parameter perturbations.

Frequency-locked breathers give rise to wide and dense RF combs which are not constrained by the length of the laser cavity. We have shown an example of a comb where the density of the spectral lines is increased by a factor of 41 compared with stationary single-pulse mode locking, thus enabling a line spacing in the sub-MHz range. Another example of a dense comb (where the density increase factor is 35) is given in Supplementary Fig. 4. Therefore, representing

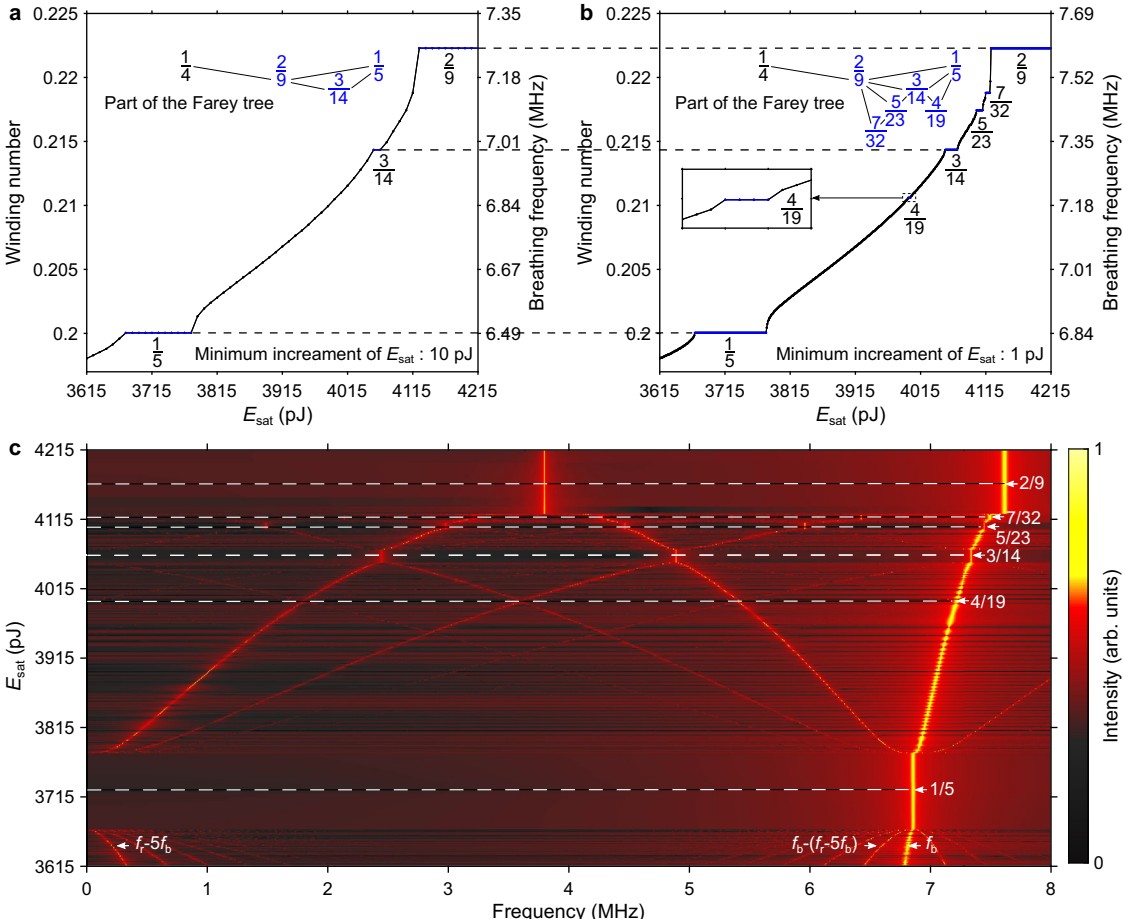

**Fig. 6 | Farey tree and devil's staircase observed in the numerical simulations.**
**a**, **b** Breathing frequency (winding number) as a function of the gain saturation
energy $E_{sat}$ (related to the pump power in the experiment) varied with a step of 10
and 1 pJ, respectively. With the smaller step, more plateaux are observed, eviden-
cing a fractal pattern. In the insets are shown the parts of the Farey tree containing

the observed Farey fractions. Since the plateau representing winding number 4/19
is very narrow, it is magnified in the inset in panel (**b**). **c** Map of spectral intensity in
the space of radiofrequency and gain saturation energy, showing the build-up of
rational winding numbers.

an alternative to fibre cavities of hundreds of metres which are
regarded as being highly unstable, controlled frequency-locked
breather lasers are attractive for many applications, for instance, in
high-resolution spectroscopy.

We note that subharmonic entrainment of breather oscillations to
the cavity repetition rate in a fibre laser was recently reported and
explained as arising between the exceptional points of a non-
Hermitian system involving two coupled modes with different
detunings[26]. However, in light of the results presented in this paper, we
believe that the observed dynamics of subharmonically entrained
breathers fall outside of the exceptional point physics and can be well
understood in the framework of frequency locking of a nonlinear
system with two competing frequencies.

Dispersion plays an important role in determining the pulse
dynamics in ultrafast fibre lasers[54–56]. The laser cavity used in this
work has a nearly zero net dispersion. We have observed that fre-
quency locking of breathers does not occur when the laser is
operated at moderate or large normal dispersion[24,49]. Thus, a very
small net cavity dispersion seems to be crucial to the emergence of
frequency-locked breathers in a fibre laser. It is worth noting in this
regard that breathing solitons at nearly zero net dispersion and
large normal dispersion differ quite significantly in respect to their
period of oscillation. Indeed, the former oscillates with a period
ranging from several to dozens of round trips while the latter
generally features a much longer period of the order of hundreds of
round trips[24], indicating that the underlying formation mechanism

could be different. Future work will thoroughly investigate the
connection between the frequency-locking mechanism and the
cavity dispersion.

## Methods
### Farey tree
The Farey tree refers to a particular sequence of rational numbers
by using the Farey-sum or median operation $\oplus$ to two adjacent
fractions, $m/n$ and $p/q$, returning a new fraction in the next lower
level of the tree by summing the denominators and numerators
separately:

$$\frac{m}{n} \oplus \frac{p}{q} = \frac{m+p}{n+q} \tag{1}$$

The physically motivated hypothesis invoked to explain the
local ordering of the hierarchy of (two-frequency) resonances is
that the larger the denominator, the smaller the plateau. The Farey
fraction or Farey mediant is the fraction with smallest denominator
between $m/n$ and $p/q$, if they are sufficiently close that $|np - mq| = 1$
– when they are called adjacents—hence it is the most important
resonance in the interval. The Farey tree provides a qualitative local
ordering of two-frequency resonances and gives rise to a curve
containing an infinite number of plateaux exhibiting self-similarity,
which is known as the devil's staircase. A detailed review can be
found in ref. 10.

## Fractal dimension of a complementary set

We have employed the equation[57]

$$\sum_i (S_i/S)^D = 1 \tag{2}$$

for computation of the fractal dimension $D$ of the Cantor set complementary (on the pump-power axis) to a complete devil's staircase. In this equation, $S$ refers to the gap (in pump power) between two parental stairs representing winding numbers $m/n$ and $p/q$, and $S_i$ corresponds to the gaps between the filial stair $(m+p)/(n+q)$ and the parental stairs.

## Evolutionary algorithm

The principle of the EA, as depicted in Fig. 1b, imitates Darwin's evolution theory: only the fittest individuals in a population survive through successive generations[58]. Here, an individual refers to a laser regime, related to the transfer function of NPE determined by the three control voltages added to the LCs; the three voltages are thus the individuals' genes. The algorithm starts with a group of individuals (population), each constituted by a collection of random genes. The output of the system is monitored for all the individuals in each generation, calculated by a user-defined merit function and a score is returned. Then the EA produces the new generation by breeding individuals from the last generation; the possibility of an individual to be chosen as a "parent" depends on their score ("roulette wheel" selection[58]). To generate two new individuals (children), the genes are interchanged between two parents who are randomly chosen. Mutation is also considered, allowing refreshing the genetic sequence. This procedure continues until the algorithm converges, giving rise to the fittest individual. In the experiment, the algorithm starts with a population containing 50 individuals and the number of individuals of the next generations is maintained at a constant value of 30 (6 parents and 24 children). It takes 2.5 min for the system to evaluate the merits of the 30 individuals in a generation.

The merit function plays a key role in realising self-tuning of the laser, which must give a larger value as the target state is approached. In the present work, we have defined and tested the following merit function for the auto-setting of an optimised self-starting frequency-locked breather regime:

$$F_{merit} = \alpha F_{ml} + \beta F_b + \gamma F_{snr} \tag{3}$$

with $F_{ml}$ being the merit function associated with the mode-locking state of the laser[59]:

$$F_{ml} = \sum_{i=1}^{i=N} I_i/N, I_i = \begin{cases} I_i, & (I_i \geq I_{th}) \\ 0, & (I_i < I_{th}) \end{cases} \tag{4}$$

where $N$ is the number of the outputting laser intensity peaks ($N = 2^{24}$, referring to a time trace of 7174 round trips), $I_i$ is the intensity at the position $I$ in the time trace and $I_{th}$ is an intensity threshold higher than the noise. Thus, $F_{ml}$ is the average intensities of the pulses and is used to exclude laser modes, including noise-like pulsing and relaxation oscillations, which can display similar RF spectral features to the breather regime. The second term $F_b$ is a merit function that distinguishes between breathers and stationary mode-locked regime, derived from the feature that the breathing frequency $f_b$ emerges as two symmetrical sidebands $f_{\pm 1}$ around the cavity repetition frequency $f_r$ as measured by the ESA ($f_b = |f_{\pm 1} - f_r|$). There are no sidebands when the laser operates at a stable mode-locked state. As a result, $F_b$ is designed to exploit the intensity ratio between the cavity repetition

rate ($f_r$) and its sidebands at $f_{\pm 1}$:

$$F_b = 1 - \sum_{f=f_r-\Delta}^{f=f_r+\Delta} I(f) / \sum_{f=f_{-1}}^{f=f_{+1}} I(f) \tag{5}$$

where the numerator and denominator in the fraction are the intensities measured across the frequency range with a width of $2\Delta$ centred at $f_r$ and the frequency span from $f_{-1}$ to $f_{+1}$, respectively. Therefore, if $F_b$ is close to zero, it represents that $f_r$ dominates and no sidebands are present, indicating that the laser works in a stationary mode-locked regime. In contrast, a far from zero value of $F_b$ means the generation of strong sidebands in the RF spectra, evidencing that breathers could be formed in the laser. In the optimisation process, the RF spectrum is calculated through fast Fourier transform of the laser intensity recorded by the oscilloscope. The weighted sum of $F_{ml}$ and $F_b$ can be used to target the regime of breather mode locking[49]. The third term in Eq. (3) is a new merit function used to distinguish between frequency-locked and unlocked breather oscillations by evaluating the strongest breathing frequency in the interval $[f_r + \delta, 3f_r/2]$:

$$F_{snr} = \max I(f), f \in [f_r + \delta, 3f_r/2] \tag{6}$$

where the frequency shift $\delta$ is used to exclude the fundamental frequency from the evaluation interval, and $f_r/2$ represents the maximum possible breathing frequency. The weights of the three components in Eq. (3) are determined empirically and set to $\alpha = 2000$, $\beta = 200$ and $\gamma = 200$.

## Numerical modelling

The generalised nonlinear Schrödinger equation is used to model the pulse evolution dynamics in optical fibres and its scalar version is[50]:

$$\psi_z = -\frac{i\beta_2}{2}\psi_{tt} + i\gamma|\psi|^2\psi + \frac{g}{2}\left(\psi + \frac{1}{\Omega^2}\psi_{tt}\right) \tag{7}$$

where $\psi = \psi(z,t)$ is the slowly varying electric field, $z$ is the propagation coordinate, $\beta_2$ and $\gamma$ are the second-order dispersion and Kerr nonlinearity coefficients, respectively, and the dissipative terms represent linear gain as well as a parabolic approximation to the gain profile with the bandwidth $\Omega$. The gain is saturated according to $g(z) = g_0\exp(-E_p/E_{sat})$, where $g_0$ is the small-signal gain, which is non-zero only for the gain fibre, $E_p(z) = \int dt|\psi|^2$ is the pulse energy, and $E_{sat}$ is the gain saturation energy determined by the pump power. The NPE-based mode-locking method can be modelled by an instantaneous and monotonous nonlinear transfer function for the field amplitude:

$$T = \sqrt{1 - q_0 + q_m / \left[1 + \frac{P(t)}{P_{sat}}\right]} \tag{8}$$

where $q_0$ is the unsaturated loss due to the absorber, $q_m$ is the saturable loss (modulation depth), $P(z,t) = |\psi(z,t)|^2$ is the instantaneous pulse power, and $P_{sat}$ is the saturation power. Linear losses are imposed after the passive fibre segments, which summarise intrinsic losses and output coupling. The numerical model is solved with a standard symmetric split-step propagation algorithm and uses similar parameters to the nominal or estimated experimental values (see Supplementary Table 1).

## Data availability

The data generated in this study have been deposited in the Zenodo database [https://zenodo.org/record/7009100#.Yv82ubdBxD8t].

## Code availability

The code that supports the findings of this study is available from the corresponding author on request.

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

## Acknowledgements

We acknowledge support from the National Natural Science Fund of China (11621404, 11561121003, 11727812, 61775059, 12074122, 62022033 and 11704123), Shanghai Municipal Science and Technology Major Project (2019SHZDZX01-ZX05), Key Project of Shanghai Education Commission (2017-01-07-00-05-E00021), National Key Laboratory Foundation of China (6142411196307), Shanghai Rising-Star Program and Science and Technology Innovation Program of Basic Science Foundation of Shanghai (18JC1412000), the Agence Nationale de la Recherche (ANR-20-CE30-004).

## Author contributions

J.P. and H.Z. initiated the work. X.W., J.P. and H.Z. performed the experiments. X.W., Y.Z., J.P. and S.B. carried out the numerical simulations. J.P., S.B., C.F and H.Z. supervised and guided the work. All authors contributed to data analysis and the writing of the paper.

## Competing interests

The authors declare no competing interests.
