## [Peer review file · Nature Communications]

REVIEWER COMMENTS

Reviewer #1 (Remarks to the Author):

The manuscript “Farey tree and devil’s staircase of frequency-locked breathers in ultrafast laser” by Wu et al. reports on the breathing frequency locking of the breathing soliton trains to the repetition rate of the mode-locked fiber ring laser at Farey fractions. To facilitate the search of the frequency-locked breathers the authors used an evolutionary algorithm and managed to demonstrate the hierarchy of different Farey fractions forming a so-called “devil’s staircase”.

The manuscript is well written and solid. The authors clearly demonstrate distinctions between the frequency-locked and frequency-unlocked breathers observed in the fiber ring laser assembly, as well as achieve the experimental generation of frequency-locked breathers with different winding numbers. While there were indeed several rather old works on the demonstration of subharmonic entrainment in externally modulated laser systems, I recognize the novelty demonstrated in the manuscript, where the locking happens between two “internal” frequencies of the nonlinear system.

My only concern about the manuscript is the very “observatory” nature of the results. As far as I can see, the pulse propagation simulations in such a system should not be a challenging task, as well as exploring the parameter space for searching the frequency-locked breathers. In my opinion, it could be great to verify the results in simulations and confirm the hypothesis about the dispersion impact on the presence of frequency locking.

Another question I would like to clarify is how reproducible the frequency-locked states when the system is being turned off and back on again? Does one need to run EA each time the system is being refreshed? Are winding numbers being reproduced?

A final comment is on the application side – I may agree with the authors that such breathing solitons with fractional numbers can be used for increasing the density of the generated comb states at the expense of introducing some additional system complexity in terms of looking for fine parameter settings, but still tend to consider the results as a fundamental phenomenon, which is nevertheless interesting.

I would support the publication of this manuscript in the Nature Communications as is, but also may suggest authors consider at least trying to provide more insights into the system behavior with simulations.

Reviewer #3 (Remarks to the Author):

this paper presents an idea of using genetic algorithm for automatically tuning the breathing frequency to analyzing the underlying nonlinear dynamics. The paper is unfortunately poorly written. The contribution of this paper is ambiguous. Fig 1. is not informative about the key concept of this manuscript. Thus I cannot recommend this paper for publication on nature communications.

Point-by-point response to the issues raised by Referees

Reviewer #1 (Remarks to the Author)

Comment 1: The manuscript “Farey tree and devil’s staircase of frequency-locked breathers in ultrafast laser” by Wu et al. reports on the breathing frequency locking of the breathing soliton trains to the repetition rate of the mode-locked fiber ring laser at Farey fractions. To facilitate the search of the frequency-locked breathers the authors used an evolutionary algorithm and managed to demonstrate the hierarchy of different Farey fractions forming a so-called “devil’s staircase”.

The manuscript is well written and solid. The authors clearly demonstrate distinctions between the frequency-locked and frequency-unlocked breathers observed in the fiber ring laser assembly, as well as achieve the experimental generation of frequency-locked breathers with different winding numbers. While there were indeed several rather old works on the demonstration of subharmonic entrainment in externally modulated laser systems, I recognize the novelty demonstrated in the manuscript, where the locking happens between two “internal” frequencies of the nonlinear system.

Reply: We are glad that the Reviewer appreciated the technical and scientific content and novelty of our work, and we thank them for providing very useful comments. It is indeed interesting that in a breathing soliton mode-locked laser an external driver is not needed, and frequency locking occurs between two “internal” frequencies. In contrast to an externally modulated system where the modulation frequency can be arbitrarily set, hence the frequency-locked states expected according to the Farey tree can be easily accessed, in a breather laser the breathing frequency is established once the laser is fabricated, while it still features a Farey tree ordering with a dimension that agrees well with that of a complete devil’s staircase.

Comment 2: My only concern about the manuscript is the very “observatory” nature of the results. As far as I can see, the pulse propagation simulations in such a system should not be a challenging task, as well as exploring the parameter space for searching the frequency-locked breathers. In my opinion, it could be great to verify the results in simulations and confirm the hypothesis about the dispersion impact on the presence of frequency locking.

Reply: We thank the Reviewer for this important remark. Various models are used in the fibre laser community depending on the level of realism that is expected. The first class of models are based on the complex Ginzburg-Landau equation (CGLE) and describe the average pulse evolution in the laser cavity with a single equation that includes normalised parameters. These master models have the advantage to be simple to handle and proved very useful in

the past to qualitatively demonstrate the existence of various nonlinear wave solutions and identify relevant parameter regions. However, these models cannot be used to make quantitative predictions about the laser behaviour, and the connection between the actual parameters of a real system and the normalised parameters of the CGLE is also quite tricky. More realistic models model each part of the laser cavity separately and use a generalised nonlinear Schrödinger equation to describe propagation in the fibre sections. Discrete models are now widely employed in the fibre laser community and have shown to be able to reproduce the main trends observed experimentally in a wide range of configurations. We have used a scalar-field discrete model to simulate the laser as described in the “Methods” section of the revised manuscript. The physical parameters of the model are close to their nominal or estimated experimental values, with the understanding that a fair evaluation of parameters such as the small-signal gain or the gain saturation energy, which are not directly accessible in the experiment, is rather difficult.

The gain saturation energy in the model is related to the pump power in the experiment, therefore we have studied the variation of the breathing frequency with the gain saturation energy. Remarkably, the simulation results - shown in Fig. 6 of the revised manuscript and included below - agree well with the experimental results shown in Fig. 5. We would like to emphasise that the simulation can reproduce the same part of the Farey tree from a breathing frequency of $1/5$ to $2/9$ as that observed in the experiment. The gaps (in gain saturation energy) between the stairs (corresponding to quasi-periodic breather states) also resemble those (in pump power) found in the experiment. The fractal dimension of the set of gaps calculated from the model is 0.873 ± 0.09 , which agrees with the experimentally calculated value and approaches that expected from a complete devil’s staircase.

Fig. 6. Farey tree and devil's staircase observed in the numerical simulations. (a,b) Breathing frequency (winding number) as a function of the gain saturation energy (related to the pump power in the experiment) varied with a step of 10 and 1 pJ, respectively. With the smaller step size, more plateaux are observed, evidencing a fractal pattern. In the insets are shown the parts of the Farey tree containing the observed Farey fractions. Since the plateau representing winding number 4/19 is very narrow, it is magnified in the inset in panel (b). (c) Map of spectral intensity in the space of radiofrequency and gain saturation energy, showing the build-up of rational winding numbers.

As pointed out by the Reviewer, numerical simulations provide the opportunity to explore the parameter space for searching frequency-locked breathers. We did so by firstly appreciating the fractal characteristics of the distribution of winding numbers in parameter space. As seen in Fig. 6(a), when the gain saturation energy is varied with a step of 10 pJ, only three plateaux can be detected. Conversely, decreasing the step to 1 pJ leads to the emergence of more plateaux (Fig. 6(b)), thus confirming the fractal structure of the winding number distribution. This results from the existence of an infinity of rational numbers between any two rational numbers, where this property of the distribution curve has given rise to the name of devil's staircase.

We have developed our numerical analysis further. In the Supplementary Information of the original manuscript, we have shown examples of different parts of the Farey tree that can be obtained experimentally by slightly changing

the initial polarisation state of the laser. Changing the polarisation state means changing the linear intracavity loss. We have confirmed this experimental observation by simulations and added a new figure (Fig. S6) in the Supplementary Information summarising the results. This figure is also included below.

Fig. S6. Farey tree and devil's staircase observed in the numerical simulations by increasing the linear intracavity loss by 1.37% above the value used to obtain Fig. 6 in the main body of the manuscript. (a) Breather frequency (winding number) as a function of the gain saturation energy (related to the pump power in the experiment). In the inset is shown the part of the Farey tree containing the observed Farey fractions. The dimension of the set complementary to the stairs is calculated to be 0.81 ± 0.03 . (b) Map of spectral intensity in the space of radiofrequency and gain saturation energy, showing the build-up of rational winding numbers. These numerical results relating to the experimental results shown in Fig. S5, confirm that a small change in the laser's initial polarisation state triggers Farey fractions belonging to a different part of the Farey tree. The simulations reveal more plateaux than those observed experimentally since the gain saturation energy can be varied with an arbitrarily small step in the model.

A further important observation emerging from our numerical analysis is that frequency locking of the breather oscillations exists also at lower levels of nonlinearity in the fibre cavity. We have indeed set the nonlinearity parameter of the standard mono-mode fibre (SMF) to half of the typical SMF28 value, and

still we have been able to observe frequency locking.

Our experiments indicate that frequency locking phenomena disappear when the laser operates at large net normal dispersion. We have performed a very large number of numerical simulations to confirm this point, and indeed we have not observed the generation of frequency-locked breathers within the range of parameters used. It is worth noting in this regard that breathing solitons at nearly zero net cavity dispersion and at large normal dispersion differ quite significantly in respect to their period of oscillation. Indeed, the former oscillate with a period ranging from several to dozens of round trips (short-period breathing) while the latter generally feature a much longer period of the order of hundreds of round trips (long-period breathing), indicating that the underlying formation mechanism could be different, and perhaps we may reach the limits of realism of the discrete model of the laser that we have used. Furthermore, other cavity parameters such as the length of the various elements might also have an impact. The space of parameters is so large that, although we feel quite confident in our statement, we cannot definitively conclude at this stage that frequency locking is fully impossible in a fibre laser with large average dispersion. This will be the subject of a future dedicated theoretical study. With the potential help of other scientists, our goal will be to fully understand the physical link that may exist between the frequency locking mechanism and the dispersion experienced by the pulses during each cavity traversal. Furthermore, we plan also to investigate the impact of the spontaneous noise from the amplification stage on the spectral purity of the frequency-locked states. As frequency locking emerges as a very new subject in ultrafast fibre lasers, we anticipate that our work will really stimulate further theoretical investigations along this line.

Comment 3: Another question I would like to clarify is how reproducible the frequency-locked states when the system is being turned off and back on again? Does one need to run EA each time the system is being refreshed? Are winding numbers being reproduced?

Reply: This is again a good point. The frequency-locked states are not self-starting meaning that if the pump power is turned off when the laser operates in a frequency-locked state and then it is turned back on again, the laser does not return to that state instantaneously. To restore the frequency-locked state, one can run the EA again, which will quickly converge to the desired state. Having performed many such experimental tests, we can claim that the frequency-locked states are reproducible. It is worth noting in this respect that many mode-locking regimes of lasers are not self-starting. An example is Ti:sapphire lasers that require an external intervention, e.g., by knocking on a component of the laser or vibrating a mirror, to make the laser emit ultrashort pulses.

Comment 4: A final comment is on the application side – I may agree with the authors that such breathing solitons with fractional numbers can be used for increasing the density of the generated comb states at the expense of introducing some additional system complexity in terms of looking for fine parameter settings, but still tend to consider the results as a fundamental phenomenon, which is nevertheless interesting.

Reply: We are glad that the Reviewer think our work is interesting from the fundamental nonlinear science standpoint. We believe that both fundamental and practical aspects should be explored in parallel in the future. As elicited by the Reviewer and described in our reply to Comment 2, there are still fundamental challenges to address, such as the connection between cavity dispersion and frequency locking. The increase in density of the generated radiofrequency (RF) combs appears also exciting for us, where establishing what is the minimum line spacing achievable is in fact related to the theoretical challenge mentioned above. Indeed, as large normal average dispersion breather lasers feature a longer breathing period than their nearly-zero dispersion counterparts, they would generate even more packed RF spectral lines should frequency locking of such lasers be possible.

I would support the publication of this manuscript in the Nature Communications as is, but also may suggest authors consider at least trying to provide more insights into the system behavior with simulations.

Reply: We are very pleased that the Reviewer supports the publication of our work as is. We really appreciate the very helpful comments made by the Reviewer, which helped us improve the quality of our work. In particular, as suggested by the Reviewer, we have performed numerical simulations of the laser to get an insight into the observed dynamics, as detailed in our reply to Comment 2. These simulations strongly confirm our experimental findings.

Reviewer #3 (Remarks to the Author):

this paper presents an idea of using genetic algorithm for automatically tuning the breathing frequency to analyzing the underlying nonlinear dynamics. The paper is unfortunately poorly written. The contribution of this paper is ambiguous. Fig 1. is not informative about the key concept of this manuscript. Thus I cannot recommend this paper for publication on nature communications.

Reply: We thank the Reviewer for their time in reviewing our manuscript. We are afraid that it is quite difficult to address comments on the unsuitability of our manuscript for publication in Nature Communications that are not supported by specific arguments and/or evidence. Nevertheless, we will try.

Figure 1(a) shows the experimental setup of the laser, and Fig. 1(b) illustrates the principle of the evolutionary algorithm used, which we believe is helpful for the readers.

We modestly think that the manuscript is well written and solid as commented by the other Reviewer who thinks that the work could be published in Nature Communications as is. Besides, we also included numerical simulations of the laser model in the revised manuscript, which further strengthen our experimental observations.

As the Reviewer may have missed the main achievements of our work – their only technical comment is about Fig. 1 -, we would like here to summarise again the significance of our work. Our experiments and simulations provide further evidence that frequency locking, which has been widely investigated in other physical systems (condensed-matter physics, Josephson junctions, circle maps, etc.), is a universal behaviour of nonlinear systems with two competing frequencies. In our work, for the first time we link frequency locking to the soliton physics, which may thus stimulate parallel studies in a variety of soliton-supporting systems. Breathing solitons have been extensively investigated in nonlinear optics in open-loop systems (for example, single-pass fibre systems (Nat. Photon. 12, 303, 2018; Phys. Rev. Lett. 122, 084101, 2019). However, in the absence of a frequency-locking mechanism, these breathers may suffer from instabilities originating from the noise of the input light. By contrast, in our work we study the dynamics of breathers in a closed-loop system – a laser resonator. This way, the universal frequency-locking process can be tailored through the nonlinear interaction between the cavity repetition frequency provided by the laser resonator and the breathing frequency. Ergo, frequency-locked breathers can be generated, showing excellent stability against cavity parameter perturbations. There is no doubt that parallel research will henceforth be triggered in many nonlinear systems, such as micro-resonators, optical parametric oscillators, and various ultrafast laser sources.

Whilst frequency locking phenomena have been extensively studied in systems with an external modulation, far less is known, by comparison, when the second frequency is not externally controlled and is intrinsic to the nonlinear system. Our work demonstrates for the first time that a fibre laser working in the breathing-soliton generation regime is a passive nonlinear system showing frequency locking at Farey fractions. The dimension of 0.906 determined from the measured devil's staircase indicates the universal nature of this nonlinear system. The breather mode-locked fibre laser thus may serve as a particularly simple model system for the investigation of universal nonlinear properties, considering both the relative simplicity and precision of fibre-optics experiments, the release from the requirement of an external modulation, and the availability of advanced tools for the characterisation and investigation of the laser output. We would like to reiterate here that contrarily to an externally modulated system where the modulation frequency can be arbitrarily set hence the frequency-locked states expected according to the Farey tree can be easily accessed, in

a breathing-soliton mode-locked laser the breathing frequency is established once the laser is fabricated, while it can be entrained by tuning the laser parameters and no external driver is needed. Given that external forces must be used to realise frequency locking in previously studied systems, the significance of our work is evident as our work opens the possibility of studying frequency locking inside a single oscillator without employing external frequency sources.

Apart from its fundamental interest, our work has also a potential for practical applications. We have indeed demonstrated that frequency-locked breathers can give rise to wide and dense radiofrequency (RF) combs which are not constrained by the length of the laser cavity. We have reported examples of combs where the density of the spectral lines is increased by a factor of more than 40 compared with stationary single-pulse mode locking, thus enabling a line spacing in the sub-MHz range. Therefore, representing an alternative to fibre cavities of hundreds of meters which are regarded as being highly unstable, controlled frequency-locked breather lasers are attractive for many applications requiring high spectral resolution such as, for instance, in spectroscopy.

Our findings were made possible by a combination of correct understanding of the physics behind the experimental observations, accurate RF spectral measurements and appropriate use of a machine learning approach for controlling the physical process. Importantly, in the revised version of the manuscript, we have performed numerical simulations of the laser model which agree very well with our experimental results. We therefore believe that our work represents an advance whose significance and impact encompass the broad field of nonlinear systems, soliton physics and practical applications, as such, it fits well the scope of Nature Communications.

In light of the above considerations, we would appreciate if the Reviewer could kindly reconsider their opinion about our work.

Yours Sincerely,

Junsong Peng, Xiuqi Wu, Ying Zhang, Sonia Boscolo, Christophe Finot, Heping Zeng

REVIEWERS' COMMENTS

Reviewer #1 (Remarks to the Author):

I thank the authors for the revision of the manuscript and addressing my comments. In my opinion the changes have significantly improved the quality of the manuscript, especially with the addition of the simulation results confirming the authors' observations.

I have no other comments and suggest to publish the manuscript without further revisions.